# Challenges in the Diagnosis of Cutaneous Mastocytosis

**DOI:** 10.3390/diagnostics14020161

**Published:** 2024-01-11

**Authors:** Knut Brockow, Rebekka Karolin Bent, Simon Schneider, Sophie Spies, Katja Kranen, Benedikt Hindelang, Zsuzsanna Kurgyis, Sigurd Broesby-Olsen, Tilo Biedermann, Clive E. Grattan

**Affiliations:** 1Department of Dermatology and Allergy Biederstein, School of Medicine and Health, Technical University of Munich, 80802 Munich, Germanytilo.biedermann@tum.de (T.B.); 2Department of Dermatology and Allergy Centre, Odense University Hospital, 5000 Odense, Denmark; 3St John’s Institute of Dermatology, Guy’s Hospital, London SE1 9RT, UK

**Keywords:** mastocytosis, diagnosis, challenges, *KIT*, dermatohistopathology, mast cells, urticaria pigmentosa, MPCM

## Abstract

Background: Mastocytosis is characterized by an accumulation of clonal mast cells (MCs) in tissues such as the skin. Skin lesions in mastocytosis may be clinically subtle or heterogeneous, and giving the correct diagnosis can be difficult. Methods: This study compiles personal experiences together with relevant literature, discussing possible obstacles encountered in diagnosing skin involvement in mastocytosis and cutaneous mastocytosis (CM). Results: The nomenclature of the term “CM” is ambiguous. The WHO classification defines CM as mastocytosis solely present in the skin. However, the term is also used as a morphological description, e.g., in maculopapular cutaneous mastocytosis (MPCM). This is often seen in systemic, as well as cutaneous, mastocytosis. Typical CM manifestations (MPCM), including mastocytoma or diffuse cutaneous mastocytosis (DCM), all share a positive Darier’s sign, and can thus be clinically recognized. Nevertheless, distinguishing monomorphic versus polymorphic MPCM may be challenging, even for experienced dermatologists. Less typical clinical presentations, such as MPCM with telangiectatic erythemas (formerly called telangiectasia macularis eruptiva perstans), confluent, nodular or xanthelasmoid variants may require a skin biopsy for histopathological confirmation. Because MC numbers in CM have a large overlap to those in healthy and inflamed skin, detailed histopathological criteria to diagnose mastocytosis in MPCM are needed and have been proposed. D816V *KIT* mutational analysis in tissue is helpful for confirming the diagnosis. Biomarkers allow the prediction of the course of CM into regression or evolution of the disease. Further diagnostic measures should screen for concomitant diseases, such as malignant melanoma, and for systemic involvement. Conclusions: Whereas in typical cases the diagnosis of CM may be uncomplicated, less typical manifestations may require specific investigations for making the diagnosis and predicting its course.

## 1. Introduction

Mastocytosis is a disease characterized by the accumulation of clonal mast cells (MCs), either only in the skin (cutaneous mastocytosis, CM) and/or in extracutaneous organs such as the bone marrow (systemic mastocytosis, SM) [1,2]. Cutaneous mastocytosis is typically seen in children [3,4]. The 2022 WHO classification divided CM into three forms, all of which may present in children: maculopapular CM (MPCM, also called urticaria pigmentosa), isolated mastocytoma and diffuse CM (DCM) (Table 1) [1,2]. Adults with cutaneous lesions of mastocytosis (CLM) suffer from SM in >90% of cases and present with MPCM. Recently, some adults with mastocytosis are identified because of hymenoptera venom anaphylaxis, together with increased baseline serum tryptase levels or activating mutations in KIT, even without skin lesions [5,6]. However, in most adults, mastocytosis is still diagnosed because of their cutaneous manifestations [4]. Thus, CLM remain the presenting key sign leading to the diagnosis of mastocytosis. In this article, we describe the challenges associated with the diagnosis of CLM, drawing upon the pooled experiences of the authors, including confirmation of the diagnosis by dermatohistopathology.

## 2. Challenges in the Use of the Nomenclature for Cutaneous Mastocytosis

The term “cutaneous mastocytosis (CM)” is often used as a description of all skin lesions associated with cutaneous mast cell accumulations in mastocytosis. However, it is defined by the European Competence Network of Mastocytosis (ECNM) and adopted by the World Health Organization as a form of mastocytosis involving only the skin, but no other extracutaneous organs (Table 2) [1,2]. As the terms “cutaneous lesions of mastocytosis (CLM)” or “cutaneous involvement in mastocytosis” sound more complicated, in this manuscript we also use the term “CM” as a synonym for the morphological description of CLM. Another example of a dermatological term which has also been used both for the diagnosis as well as for the morphological picture is the term “urticaria”. “Urticaria” can be used as a description of a clinical picture characterized by wheals, as well as a diagnostic term (also including the manifestation of angioedema) [7]. According to the WHO, in children, the diagnosis of CM is assumed even without confirmation by a bone marrow biopsy (BMB), as in this age group, systemic disease is rare [1,2]. Conversely, >90–95% of adults with skin involvement suffer from SM [8]. In this age group, however, a diagnosis of SM has been recommended by ECNM to be always confirmed by a BMB by checking for major and minor criteria for SM (Table 3) [8,9]. Adults with CM, where a BMB has not been performed are diagnosed as “mastocytosis in the skin (MIS)”, which is another descriptive term used only for diagnosis and not for morphology. This may cause confusion [10]. Furthermore, the morphological description MPCM, as a synonym for “urticaria pigmentosa”, carries “CM” in its name but can also be used in patients with SM and is not meant to implicate pure cutaneous involvement. “Urticaria pigmentosa” is an older term still occasionally used for skin lesions of mastocytosis, though neither do wheals occur in nonaffected lesional skin, nor are the lesions always strongly pigmented [3]. It is still, nevertheless, widely used because of its simplicity. As these definitions do not appear self-evident but require an in-depth study of the official nomenclature for correct use, they are a challenge.

## 3. Clinical Diagnosis of Cutaneous Mastocytosis

### 3.1. Typical Clinical Presentations

Typical cutaneous lesions of mastocytosis can be recognized clinically by experienced dermatologists. In childhood, MPCM is most common, but may present with different morphological features [4]. Onset is usually within the first 6 months of life, but it may already be present at birth. Most children display a variant with red-brown, often round or oval, flat macules of different sizes, mostly with sharp margins (Figure 1a). However, other children show elevated plaques or nodules (Figure 1b). Lesions in children are of variable shape and size and larger than those found in patients with adulthood-onset mastocytosis [11]. They are often more brownish than red, asymmetric, typically involving the head and neck in addition to the trunk and extremities. These lesions are mostly oval, but they may present with different shapes. Thus, the clinical pictures seen in childhood MPCM have been termed the “polymorphic variant” of MPCM and are normally different from those in adults [4]. Adults with CM present with “monomorphic” MPCM (Figure 1c) [4]. Lesions consist of disseminated small (3–5 mm) red-brown macules or slightly raised papules, predominantly located on the upper thighs and trunk [4]. However, the degree of difference needed to call lesions ‘polymorphic’ remains undefined in terms of (i) shape, (ii) color and (iii) size, and it is mostly intuitive [12]. Thus, the rate of agreement was below expectations on classifying cases with monomorphic versus polymorphic CM in a study analyzing pictures of 19 sample cases with childhood mastocytosis, with just a ‘fair’ interobserver variability for 10 experts and ‘slight’ variability among 129 dermatologists [13]. 

Mastocytomas and diffuse cutaneous mastocytosis (DCM) are further manifestations of childhood-onset CM [4,14]. Solitary or up to three mastocytomas are common and are often present at birth. These lesions are mostly mildly elevated, well demarcated, solitary yellowish red-brown plaques or nodules, typically 0.5–4 cm in diameter (Figure 1d). DCM is a rare disorder characterized by diffuse mast cell infiltration of large areas of the skin that presents in the first year of life. Severe edema and leathery indurations of the skin result in an accentuation of skin folds (pseudo-lichenified skin) and a peau-d’orange-like appearance. Infants and young children with considerable mast cell infiltration of the skin sometimes exhibit blister formation in the first three years of life [3]. Blisters can occur in the first three years of life on more indurated lesions, particularly on mastocytomas or in DCM. Typical skin symptoms associated with CM are pruritus, whealing of MPCM skin lesions and flushing.

### 3.2. Darier’s Sign

When mechanically irritated, mastocytosis skin lesions may develop an edematous wheal (Figure 1d). This reaction is referred to as Darier’s sign. Darier’s sign can be elicited by stroking a mastocytosis skin lesion with a blunt object, such as a wooden tongue depressor or a calibrated dermatographometer (HTZ, East Grinstead, UK) set at 20 and 36 g/mm^2^. A strong positive Darier’s sign is often found in pediatric patients and in patients with more indurated lesions. In those, Darier’s sign should be elicited with caution, as flushing or blister formation may occur. A classical wheal limited to CM skin is pathognomonic for mastocytosis, whereas extension of the wheal to uninvolved skin also indicates symptomatic dermographism (urticaria factitia) [7]. However, Darier’s sign is not always positive in CM patients, particularly in those with mild macular lesions or in patients who have taken antihistamines [3]. 

### 3.3. Less Typical Clinical Presentations

Some patients only present with very few inconspicuous, hardly visible skin lesions of CM, which may be overlooked (“occult mastocytosis”) until they develop other sings, such as hymenoptera venom anaphylaxis, leading to the final diagnosis of mastocytosis (Figure 2a) [15]. These cases may present as a diagnostic challenge. Furthermore, telangiectasia macularis eruptiva perstans (TMEP) was previously considered as a separate form of CM (Figure 2b) [16,17]. However, mastocytosis patients with telangiectatic skin lesions typically also show small, typical maculopapular skin lesions, which are sufficient for diagnosis. In the new classification, TMEP is no longer listed as a subform [1,2,4,18]. MPCM lesions may coalesce if they are numerous. This may lead to adult patients presenting with a high degree of confluence of MPCM, resulting in large red-brown or sometimes livid diffuse skin lesions (Figure 2c) [19]. In other patients, red-brown or livid nodules may be found. Sometimes, these nodules may lead to xanthelasmoid cutaneous involvement, characterized by yellowish skin changes [20,21]. Extensive blistering in young children may indicate severe mast-cell-mediated complications with severe skin involvement, such as DCM, but CM in neonates is also a differential diagnosis for neonates developing multiple blisters [22,23]. 

### 3.4. Differential Diagnoses

The diagnosis of CM is based on clinical findings, but it is more often delayed due to lack of clinical awareness of the disease than by a wrong diagnosis. Differential diagnoses, which may resemble less typical CM, are listed in Table 4. In adults with MPCM, monomorphic disseminated symmetrical lesions on the thighs and trunk are quite typical, but similar-looking lesions can also occur in other dermatological diseases. Particularly, lentigines, pigmented nevi, essential telangiectasias, pityriasis lichenoides chronica, neurofibromatosis or lichen planus exanthematicus may resemble adult MPCM. Many patients previously diagnosed with TMEP probably have essential telangiectasias (Figure 2d). The fact that authors have proposed solely clinical presence of matted telangiectasias as the criterium for TMEP, without the need of mast cell accumulation in the dermis and without a positive Darier’s sign, has led to an overdiagnosis of TMEP. As in children with MPCM, the macroscopic picture is more heterogeneous and the list of differential diagnoses is even larger. It may include pigmented dermatological diseases, such as postinflammatory hyperpigmentation, atrophodermia idiopathica, neurofibromatosis, histiocytosis, xanthomas or chronic urticaria and pityriasis versicolor. Additionally, parapsoriasis en petites plaques and fixed drug eruption may be observed in older children. In children, juvenile xanthogranuloma or smooth muscle hamartoma are often misdiagnosed as mastocytoma, and vice versa. Bullous eruptions in young children may occur in DCM and occasionally in very young children with widespread MPCM, but they may also manifest in children with epidermolysis bullosa, bullous impetigo, linear IgA disease or early incontinentia pigmenti [23]. 

## 4. Confirmation of the Diagnosis of Cutaneous Mastocytosis

For dermatologists, typical skin lesions together with a clearly positive Darier’s sign and clinical exclusion of differential diagnoses are sufficient to confirm the diagnosis of CM [4]. In children with indurated skin lesions not taking anti-allergic drugs, Darier’s sign should test positive. It has to be noted, however, that lesions of differential diagnoses may also show mild erythema and minimal edema upon strong mechanical stroking, which has been named pseudo-Darier’s sign [24,25]. Darier’s sign is only pathognomonic, if strongly positive, e.g., in mastocytoma, DCM or MPCM with dense MC aggregates. On the other hand, a negative Darier’s sign does not rule out CM, particularly in low-grade stable adult disease [26,27].

### 4.1. Dermatohistopathology

If CM is atypical, Darier’s sign is negative, or other dermatological diseases are included in the differential diagnosis, dermatohistopathology is required for diagnosis [4,10]. Biopsies show increased numbers of MC in lesional skin [28,29,30]. Mast cells can be visualized via Giemsa or toluidine stains, but even better through antibody staining against tryptase or CD117 [9,28,31]. Increased numbers of mast cells in biopsy sections of lesional skin and activating *KIT* mutation detection in lesional skin tissue have been proposed as minor criteria for CM [10]. However, there is no fixed cutoff at which MC counts can be said to be definitely increased. Only a few laboratories are able to sequence *KIT* from skin tissue at present. In mastocytomas and DCM, typical clinical skin lesions of mastocytosis in combination with a strongly positive Darier’s sign confirm the diagnosis, and unique histopathological criteria exist (very high MC counts, MC aggregates >15, cuboid CD117^+^ and tryptase^+^ cells) [28,32]. In MPCM, the number of tissue MC in CM is variable and may overlap with MC numbers in healthy skin or in inflammatory skin diseases. Dermatohistopathology may return a nondefinitive result of “compatible with mastocytosis” because MCs are normal resident cells of the human dermis, and the increase in MC numbers in subtle MPCM may not be different from that in the skin of inflammatory dermatoses, particularly in atopic dermatitis, urticaria and pruritus (Figure 3) [30].

Recently, detailed histopathological criteria to diagnose mastocytosis in MPCM have been proposed [28]. An MC density of >27 nucleated MC per high-power field (HPF) has a specificity of 97% and a sensitivity of 78%. However, many CM patients do not have such a high MC density. It was hypothesized that the composition of the cellular infiltrate might be able to differentiate between CM and other diseases. Surprisingly, a CD3^+^ T-cell infiltrate was not only present in inflammatory diseases, but also in most patients with CM. The CD3^+^ T-cell density did not differ between mastocytosis and controls. A moderate MC infiltration >12 MC/HPF was indicative for CM (specificity 92%, sensitivity 91%), but further histopathological signs for mastocytosis should be looked for: MC clusters with >3 nucleated MC, interstitially located MC and increased pigmentation of the basal epidermis in Caucasians. In centers where the *KIT* D816V mutational analysis is not available, a scoring model to predict MPCM with major and minor criteria has been proposed in Figure 4.

### 4.2. KIT Mutation Analysis

Detection of a *KIT* mutation at codon 816 in skin tissue is sometimes required as a specific and sensitive method for making the diagnosis of CM [33]. In one study, specificity of detection of *KIT* D816V in lesional skin tissue in patients with adult MPCM was 100%, with a sensitivity of 87.5% [28]. In centers where detection of *KIT* D816V in lesional skin tissue is not possible, the detection of this mutation in peripheral blood may be sufficient to confirm the diagnosis [34]. The presence of this mutation in peripheral blood only represents one of four minor criteria for SM, whereas *KIT* D816V was not identified in the peripheral blood of children known to have only cutaneous disease [35].

## 5. Prediction of Evolution or Resolution of Disease

It is discussed among experts that children with mastocytomas show complete resolution in childhood. About half to two thirds of children with MPCM have spontaneous resolution before adulthood and most others show a regression of skin lesions or a stable disease [36,37]. This is paralleled by a decrease of increased baseline serum tryptase (BST) levels [11,38]. Children with SM or presence of the D816V mutation have a worse prognosis for clearance of skin lesions [38]. It has been reported that those children with small lesions resembling adult-type MPCM may show a continuation of the disease into adulthood [39]. Other predictive factors for the counselling of parents remain unknown. 

The course of mastocytosis in adults is usually chronic [5]. After development, skin lesions, mast cell load mirrored by BST and sometimes clinical symptoms appear to increase over the next years or decades [11]. This is followed in most patients by a stable disease phase. The implication of regression of skin lesions for adult patients remains controversial. It has been associated with the development of advanced SM, and it is speculated that in these patients, abnormal neoplastic MCs may lose homing receptors for skin tissues [40]. However, in 10% of patients over 50 years of age, CM may regress or even disappear in parallel with biological regression of clinical symptoms and mast cell load, although in the bone marrow, MC aggregates are still present [41]. In adults with CM and cutaneous or indolent systemic disease, progression into an aggressive advanced form of disease is seldom. The cumulative probability of disease progression from ISM into advanced SM was calculated to be 1.7% in ten years [42]. Serum β2-microglobulin in combination with the presence of *KIT*-mutation in all hematopoietic lineages were good parameters for predicting disease progression [42]. Patients developing pronounced confluence of livid MPCM lesions may develop smoldering SM with a high MC burden, BST > 200 ng/mL, and organomegaly. These patients may have a higher risk of disease progression [43].

## 6. Further Assessments in Patients with Cutaneous Mastocytosis

In patients with CM, physical examination, assessment of skin lesions, serum tryptase determination and a complete blood count should be done at yearly intervals. A full skin examination in CM is advisable, as patients with CM have been found to have a higher risk for developing skin cancer, particularly malignant melanoma and nonmelanoma skin cancer [44,45,46].

Mast cell mediator-related symptoms may significantly affect the quality of life [47,48]. Cutaneous and gastrointestinal complaints, anaphylaxis, musculoskeletal pain, fatigue and osteoporosis are the main symptoms and signs to be assessed. In all adults with SM, osteodensiometry using the DXA method (±spine X-ray) with a follow-up should be performed, as the risk of osteopenia and osteoporosis is increased in SM of all subtypes, including ISM patients with few symptoms and low levels of tryptase [44,49].

It remains questionable if regular abdominal sonography is needed in patients with stable, uncomplicated indolent or pure cutaneous disease, particularly in children with CM. Furthermore, the guidelines recommend bone marrow biopsy and aspiration in all adult patients with mastocytosis. However, the value of this examination remains unproven for those patients with stable, uncomplicated indolent or pure cutaneous disease in whom hematological involvement or aggressive disease is not suspected. It has been reported that skin lesions in adults alone can predict systemic disease without the need for a BMB [50]. Furthermore, an analysis of *KIT* mutations, especially the D816V mutation in the peripheral blood, may be performed in patients in whom the diagnosis is uncertain. In adults with moderate or severe ISM and in those with advanced forms of the disease, other laboratory parameters, particularly eosinophils, alkaline phosphatase and β2-microglobulin should be measured to determine the risk of disease progression [5,42,51].

Further examinations, such as gastrointestinal endoscopy or biopsy to obtain tissue specimens from the liver, spleen, gastrointestinal tract or lymph nodes, are usually considered only when these organs are suspected to be involved, and the biopsy would yield clinically useful information.

## 7. Conclusions

The nomenclature used for various forms of mastocytosis is complex and requires knowledge of the definitions, such as the use of “CM”, which is defined as a diagnosis only for patients without involvement of extracutaneous organs. However, the term “CM” may also be helpful for describing all skin lesions occurring in mastocytosis. Whereas confirmation of typical MPCM, mastocytoma or DCM by clinical examination and Darier’s sign is mostly straightforward, often dermatohistopathology or, rarely, even analysis of the D816V *KIT* mutation is required for diagnosis in patients with few skin lesions or atypical presentations of other dermatological differential diagnoses. Children without signs of systemic disease have a good prognosis for regression of CM. In adults, the disease remains mostly stable. Only few older patients show a clearance or marked regression of skin lesions. However, progression from CM or ISM into an advanced aggressive form of the disease is also rare.

## Figures and Tables

**Figure 1 diagnostics-14-00161-f001:**
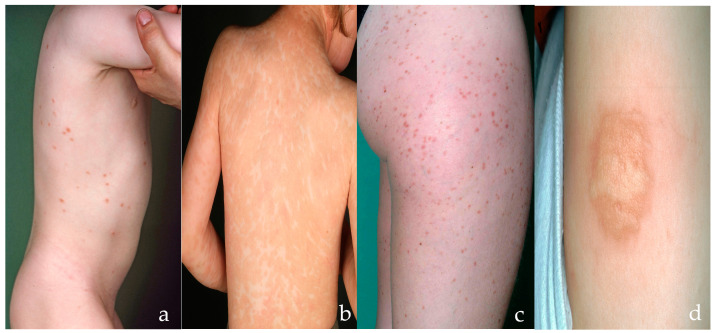
(**a**) Multiple disseminated asymmetric sharply defined flat red-brown macules in a child with childhood maculopapular cutaneous mastocytosis (MPCM). (**b**) Skin lesions may also be few or numerous with different sizes and shapes. A child with multiple disseminated light-brown plaques is shown as a presentation of MPCM. (**c**) Disseminated small red-brown macules or slightly elevated papules on the thigh typical for adult-onset MPCM. (**d**) Solitary yellowish-red-brown plaques of a mastocytoma in a child, with whealing along the stroke line of a wooden spatula only in involved skin, recognized as a positive Darier’s sign.

**Figure 2 diagnostics-14-00161-f002:**
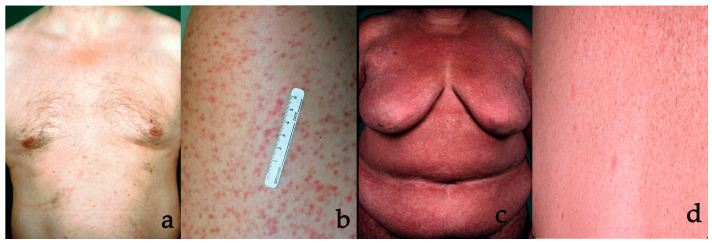
Less typical presentations of cutaneous mastocytosis (CM) and differential diagnosis. (**a**) Macular red-brown skin lesions may be few in number, sometimes even fewer than shown in this picture, which may be easily overlooked. This form has been called “occult CM”. (**b**) Lesions can be less pigmented or not pigmented and red, which, together with a matted telangiectatic appearance, has formerly been called telangiectasia macularis eruptiva perstans (TMEP). (**c**) Extensive maculopapular CM may lead to widespread, nearly confluent lesions. (**d**) Such matted telangiectasias are in fact essential telangiectasias and not TMEP, as they are lacking specific mastocytosis criteria. This patient also presents multiple lentigines, another differential diagnosis, when present in numbers and in locations typical for CM.

**Figure 3 diagnostics-14-00161-f003:**
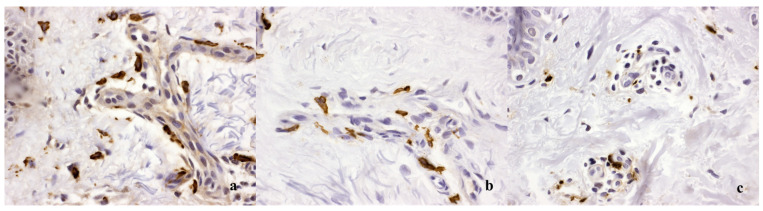
Histopathology of a lesion from a patient with cutaneous mastocytosis may be diagnostic because of (**a**) a high number of mast cells in the dermis, but it may also be nondiagnostic, (**b**) showing not significantly increased mast cell numbers in comparison to (**c**) a control patient with atopic dermatitis and may require further criteria to be looked at. Tryptase stains at 40×.

**Figure 4 diagnostics-14-00161-f004:**
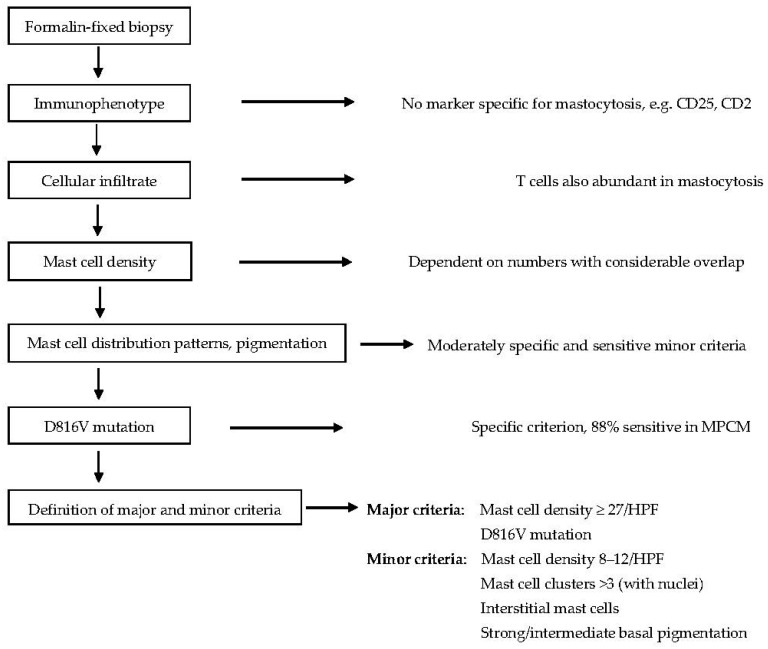
Considerations for the definition of major and minor criteria for the diagnosis of cutaneous mastocytosis in formalin-fixed skin biopsies. Presence of one major criterion, or mast cell density of ≥12 nucleated mast cells/high-power field, together with two other minor criteria, is sufficient for the diagnosis of CM.

**Table 1 diagnostics-14-00161-t001:** Classification of cutaneous mastocytosis.

Subforms
Maculopapular cutaneous mastocytosis (synonym urticaria pigmentosa) (monomorphic and polymorphic variants)
Diffuse cutaneous mastocytosis
Cutaneous mastocytoma

**Table 2 diagnostics-14-00161-t002:** The WHO classification of mastocytosis.

Form of Mastocytosis	Abbreviation	Characteristics
Cutaneous mastocytosis - Maculopapular CM - Monomorphic - Polymorphic - Diffuse CM - Cutaneous mastocytoma - Isolated - Multilocalised	CM MPCM DCM -	Only the skin is involved. Disseminated lesions. Small uniform lesions, adult type. Larger variable lesions, childhood type. Diffuse indurated skin lesions. Solitary plaques or nodules. One lesion. Up to three lesions.
Systemic mastocytosis - Bone marrow mastocytosis - Indolent systemic mastocytosis - Smoldering systemic mastocytosis - Aggressive systemic mastocytosis - Systemic mastocytosis with an associated hematological neoplasm	SM BMM ISM SSM ASM SM-AHN	Extracutaneous organs are involved. SM is present only in the bone marrow. Benign form of SM. SM with high mast cell load. Aggressive form of SM. Additional hematological neoplasm.
- Mast cell leukemia	MCL	Circulating mast cells in blood.
- Well-differentiated systemic mastocytosis	WDSM	Morphologic variant that may occur in any SM type/subtype
Mast cell sarcoma	MCS	Malignant mast cells in tissue.

**Table 3 diagnostics-14-00161-t003:** Refined major and minor criteria for systemic mastocytosis (SM) [1,8]. If at least one major and one minor or three minor criteria are fulfilled, the diagnosis is SM.

Criterion	Characteristics	Tissue
Major	Multifocal dense infiltrates of mast cells (≥15 mast cells in aggregates).	Bone marrow biopsies and/or in sections of other extracutaneous organ(s).
Minor criterion 1	≥25% of all mast cells are atypical or are spindle-shaped.	Atypical cells on bone marrow smears. Spindle-shaped mast cells in bone marrow or other extracutaneous organs.
Minor criterion 2	KIT-activating point mutation(s) at codon 816 or in other critical regions of *KIT*.	Bone marrow, blood or another extracutaneous organ.
Minor criterion 3	Mast cells express one or more of: CD2 and/or CD25 and/or CD30.	Bone marrow, blood, or another extracutaneous organ.
Minor criterion 4	Baseline tryptase >20 ng/mL. In case of an unrelated myeloid neoplasm, an elevated tryptase does not count as an SM criterion. In case of a known hereditary alpha-tryptasemia (HαT), the tryptase level should be adjusted.	Serum

**Table 4 diagnostics-14-00161-t004:** Typical differential diagnoses of adult or childhood maculopapular cutaneous mastocytosis (aMPCM, cMPCM) and mastocytoma requiring biopsy.

Differential Diagnosis	Similarity to CM	Different Aspects
Lentigines and pigmented nevi	Disseminated monomorphic small hyperpigmented lesions may mimic MPCM.	Hyperpigmentation without redness. Often solitary or regional.
Pityriasis lichenoides chronica	Multiple disseminated red small papules on trunk in children/young adults may mimic MPCM.	Very mild scaling, multiple often symmetrical lesions.
Lichen planus pigmentosus or exanthematicus	Multiple disseminated small hyperpigmented or erythematous macules and papules mimic MPCM.	Acute onset, severe itching.
Parapsoriasis en petites plaques	Disseminated red-brown pigmented small macules on lateral trunk.	Typically digital appearance, atrophic aspect, mild scaling.
Pityriasis versicolor	Hyperpigmented or erythematous macules on the trunk.	Scales after scratching, spores in microscopy, confluent lesions often on upper trunk.
Multiple café au lait spots (CALMs)	Flat disseminated brown multilocular hyperpigmented birth marks.	Ovoid or jagged borders. The presence of ≥6 lesions >5–15 mm are neurofibromatosis criteria.
Neurofibromatosis	Single or multiple firm, skin-colored nodules or tumors (neurofibromas) +/− CALMs.	Peau-d’orange aspect of mastocytomas is absent.
Atrophodermia idiopathica	Disseminated larger multilocular hyperpigmented lesions often on the trunk beginning in childhood or early adults.	Superficial form of morphea with epidermal atrophy.
Ashy dermatosis (erythema chronicum perstans)	Disseminated larger confluent gray-brown macules, different sizes at trunk, extremities.	Hyperpigmentation without redness, often larger macules.
Postinflammatory hyperpigmentation	Solitary hyperpigmented macule(s).	No itch, history of previous inflammation.
Fixed drug eruption	First red macule, possible central blister, later hyperpigmentation. Multiple lesions possible.	Often solitary, history of drug intake.
Histiocytosis X (Abt-Letter-Siwe)	In early childhood, in sweat areas (breast, back, inguinal) small, disseminated papules.	Scales, crusts, severely ill patient.
Plane warts (Verrucae planae)	Multiple pale red-brown small papules.	Localized lesions, mostly face, aggregated.
Chronic urticaria	Multiple itching red lesions.	Acute development of lesions, locations change over time, severe itching.
Juvenile xanthogranuloma	Multiple yellow-red papules and nodules in childhood.	No itch, no confluence.

## Data Availability

Access to original data is restricted due to data protection guidelines and can be granted on reasonable request.

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
