# Peer review of "Challenges in the Diagnosis of Cutaneous Mastocytosis"

_diagnostics, 2024, doi:10.3390/diagnostics14020161_

Round 1

Reviewer 1 Report

Comments and Suggestions for Authors

Very good article, well presented data.

Author Response

Thank you very much for the positive review.

Reviewer 2 Report

Comments and Suggestions for Authors

It's a very interesting paper and will have an important impact in the field of mastocytosis.

Author Response

(The authors gave the same response as above.)

Reviewer 3 Report

Comments and Suggestions for Authors

Dear Authors;

In 2022, two independent updates to the previous 2016 World Health Organization (WHO) classification and diagnostic criteria (revised 4th edition) for mastocytosis were published, namely the International Consensus Classification (ICC) of Myeloid Neoplasms and Acute Leukemias1 and the 5th edition of the WHO Classification of Haematolymphoid Tumors (Tables 1–4).

1-Arber DAOrazi AHasserjian RP, et al. International consensus classification of myeloid neoplasms and acute leukemias: integrating morphologic, clinical, and genomic dataBlood2022140(11): 1200-1228.

2-Khoury JD, Solary E, Abla O, et al. The 5th edition of the World Health Organization classification of haematolymphoid tumours: myeloid and histiocytic/dendritic neoplasms. Leukemia. 2022; 36(7): 1703-1719

This article is based on old calcification (2016). It is not appropriate to publish it in this state. It can be evaluated if it is rewritten using current classification criterias.

Author Response

Thank you very much for the helpful correction.

The two recommended references are now cited and replace the references from 2016.

Table 2 was adapted to also include bone marrow mastocytosis and mast cell sarcoma reflecting the updated classification. The sub-subforms of polymorphic and “monomorphic” MPCM and isolated and multilocalised cutaneous mastocytoma have also been added.

Reviewer 4 Report

Comments and Suggestions for Authors

In Fig 1 and Fig.2, the images seem distorted (the letters in the right corner).

Fig 1d. I admit that I have never seen such a strong and brownish Darier’s sign in my 15 years of Dermatology  practice.

Line 129 The number of the figure is missing “dermographism (urticaria factitia, Fig. )”

Comments on the Quality of English Language

Line 144 The term “nodules” is more often use instead of latin “noduli” (“Sometimes these noduli”), as it is mentioned in the same line “brown or livid nodules may be found

Author Response

Answers to the reviewer's comments:

In Fig 1 and Fig.2, the images seem distorted (the letters in the right corner).

The reviewer is correct. As I provided the figures and figure legend, but the finaly layout of the pictures of Fig. 1+2 has been done by MDPI editing. I do ask for a deletion of the letters a-d in the right hand bottom corner and reinserting them without distortion. 

Fig 1d. I admit that I have never seen such a strong and brownish Darier’s sign in my 15 years of Dermatology practice.

Thank you. The red-brownish hue is due to the mastocytoma and the white dermographism on the mastocytoma represents the edema of the Darier's sign.

Line 129 The number of the figure is missing “dermographism (urticaria factitia, Fig. )”

Thank you very much. We added the number of the Figure (Fig. 1d).

Comments on the Quality of English Language

Line 144 The term “nodules” is more often use instead of latin “noduli” (“Sometimes these noduli”), as it is mentioned in the same line “brown or livid nodules may be found

The term "noduli" has been replaced by "nodules" in line 144. We checked, there was no other mentioning of "noduli" throughout the rest of the text.